# Biopolymer Recovery from Aerobic Granular Sludge and Conventional Flocculent Sludge in Treating Industrial Wastewater: Preliminary Analysis of Different Carbon Routes for Organic Carbon Utilization

**Francesco Traina, Santo Fabio Corsino \***, **Michele Torregrossa** and **Gaspare Viviani**

Dipartimento di Ingegneria, Università degli Studi di Palermo, Viale delle Scienze Ed. 8, 90128 Palermo, Italy
* Correspondence: santofabio.corsino@unipa.it; Tel.: +39-091-2386-1929

**Abstract:** The recovery of biopolymers from sewage sludge could be a crucial step in implementing circular economy principles in wastewater treatment plants (WWTP). In this frame, the present study was aimed at evaluating the simultaneous production of polyhydroxyalkanoates (PHA) and extracellular polymeric substances (EPS) obtainable from the treatment of agro-industrial wastewater. Two biological enrichment systems, aerobic granular sludge (AGS) and a conventional activated sludge operating as a sequencing batch reactor (SBR), were monitored for 204 and 186 days, respectively. The maximum biopolymers accumulation capacity was close to 0.60 mgPHA-EPS gVSS$^{-1}$ in the AGS when operating at 3 kgCODm$^{-3}$d$^{-1}$, whereas in the SBR, it was about half (0.35 mgPHA-EPS gVSS$^{-1}$). Biopolymers extracted from the AGS were mainly constituted by EPS (>70%), whose percentage increased up to 95% with the OLR applied in the enrichment reactor. In contrast, SBR enabled obtaining a higher PHA production (50% of the biopolymers). Results suggested that organic carbon was mainly channeled toward metabolic pathways for extracellular storing in AGS, likely due to metabolic stressors (e.g., hydraulic selection pressure, shear forces) applied for promoting aerobic granulation.

**Keywords:** aerobic granular sludge; biopolymers; circular economy; extracellular polymeric substances; polyhydroxyalkanoates; sewage sludge; wastewater treatment





## 1. Introduction

During the last few years, the efforts to apply the biorefinery concept to wastewater treatment plants (WWTP) have been constantly growing. The wastewater biorefinery approach aims to overcome the view of WWTPs for remediating wastewater to an acceptable quality only to facilities aimed at enhancing energy and resource recovery [1]. According to this view, wastewater is considered a renewable resource from which water can be reclaimed, and energy and secondary raw materials can be recovered for several purposes [2].

Recent EU regulations (Directive 2018/851/EC) emphasized that overcoming the criticalities of WWTP management by exploiting environmentally-friendly solutions for resource recovery could be a valuable approach in line with sustainability and circularity concepts [3]. In this frame, one of the main criticalities in WWTP is related to sludge management and disposal since this can account for about 50 percent of total operating costs (OPEX) [4]. Therefore, the valorization of sewage sludge generated from wastewater treatment could be a sustainable solution to address both the management issues and to implement circular economy principles for the recovery of high-value products [5]. Among these, polyhydroxyalkanoates (PHAs) have seen growing interest due to their wide application as precursors to bioplastics and environmental-friendly materials [6]. PHAs are

biodegradable, biocompatible and non-toxic biopolymers synthesized by several microbial species as a carbon and energy reserve under nutrient-unbalanced conditions [7].

The current industrial production of PHA is linked to the use of pure bacterial cultures. Although this process enables obtaining a very high PHA productivity (up to 90% cell dry weight), on the other hand, it requires selected substrates, sterile conditions and a high oxygen demand [8], thereby leading to a high economic burden [9]. Using mixed bacterial cultures (MMC) from the activated sludge process is largely considered a more sustainable solution. Indeed, MMC does not require sterile conditions and allows using organic matter from wastewater as feedstock. PHA production by MMC is carried out in a three-stage process involving the production of volatile fatty acids by acidogenic fermentation of the organic feedstock (stage 1), the enrichment of the MMC with PHA-accumulating organisms (stage 2) and the intracellular accumulation of PHA (stage 3). Generally, this is obtained in a side-stream line operating in parallel with the main one dedicated to wastewater treatment. Despite the above advantages over pure microbial cultures, PHA production by MMC results in lower PHA accumulation yields [10]. Effluents generated by agro-food industries were recently exploited as secondary feedstocks to increase PHA productivity, as these are characterized by a higher organic substance content than municipal wastewater [11]. Nevertheless, recent studies indicated that the breakeven price for the process must be reduced in order to make it more attractive [12], for instance, by exploiting additional simultaneous pathways for the recovery of other resources.

To increase the cost-effectiveness of the process and maximize resource reclamation, the possibility of recovering other value-added co-products such as EPSs (extracellular polymeric substances) together with intracellular polymers was recently proposed [13]. EPSs are products of microbial metabolism, mainly composed of proteins (PS), carbohydrates (PN) and, in minor quantities, nucleic acids, lipids and various heteropolymers [14]. EPSs are functional substances that protect cells from external environmental agents such as toxic substances, and similarly to PHAs, they can be used as a source of carbon and energy under nutrient-lacking conditions [15]. EPSs represent a potential recoverable resource that can replace alginate in the pharmaceutical, food, pharmaceutical and textile industries [16,17].

The simultaneous recovery of EPS and PHA was recently investigated in the literature. For instance, Kopperi and coauthors [18] demonstrated that it is possible to achieve the simultaneous production of PHA and EPS by isolated *Providencia* sp. Similarly, EPS and PHA were successfully produced by MMC in treating municipal wastewater [19]. In this frame, a suitable technology for maximizing biopolymer recovery from wastewater treatment while implying a smaller footprint and less energy requirement than a conventional activated sludge system is aerobic granular sludge (AGS). In the AGS system, microbial adhesion of microorganisms in large particles (>1 mm) occurs, and EPS constitutes the structural elements of such bio-aggregates [20]. Pronk et al. [21] demonstrated that different types of biopolymers could be extracted from AGS depending on the operating conditions applied. Indeed, considering that the key element leading to the synthesis of PHA and EPS is carbon, it was postulated that a regulatory mechanism in carbon utilization could switch the pathway from PHA polymerization to EPS synthesis and vice versa [18,22]. However, no evidence of this was reported in previous studies. Moreover, the low organic carbon concentration in municipal wastewater could be the trigger of such a process limitation. In this respect, any other studies are available in the literature exploiting the possibility of using high-strength wastewater as organic feedstock.

From an accurate literature review to the best of the authors' knowledge, any relevant studies are available in the literature on the recovery of biopolymers (EPS+PHA) from AGS treating industrial wastewater with high-carbon content. Considering this, the present study was aimed at evaluating the simultaneous production of PHA and EPS obtainable from the treatment of agro-industrial wastewater by AGS. In more detail, the effect of the organic loading rate (OLR) on the PHA and EPS recovery yields was studied, and the results were compared with that of a conventional activated sludge plant operating in parallel with the AGS system.

## 2. Materials and Methods

### 2.1. Wastewater Characterization

The wastewater was collected from a citrus processing industry located in Palermo (Italy). Citrus wastewater was generated by the processing of different fruits such as oranges, tangerines, and lemons, as well as by the machinery washing and essential oil extraction operations. Citrus wastewater was characterized by a low pH equal to 4.5, on average, high chemical oxygen demand (COD) ranging between 4.5–5 gCOD L$^{-1}$ and unbalanced nutrients (carbon to nitrogen ratio greater than 200). Table 1 summarizes the average values of the main characteristics of wastewater.

**Table 1.** Main values of the principal characteristics of citrus wastewater.

| Parameter | Value |
| --- | --- |
| TCOD [mg L$^{-1}$] | 4486 $\pm$ 391 |
| SCOD [mg L$^{-1}$] | 3281 $\pm$ 195 |
| TN [mg L$^{-1}$] | 21 $\pm$ 7 |
| TP [mg L$^{-1}$] | 12.1 $\pm$ 4.3 |
| pH [-] | 4.2 $\pm$ 0.3 |

Legend: TCOD (Total Chemical Oxygen Demand); SCOD (Soluble Chemical Oxygen Demand); TN (Total Nitrogen); TP (Total Phosphorous).

### 2.2. Experimental Setup

The experiment was carried out using a three-stage process consisting of a fermenter (i), a sequencing aerobic granular sludge (AGS) reactor for MMC enrichment (ii) and a fed-batch reactor (FBR) (iii) to maximize the production of EPS and PHA. The bioreactor for the enrichment of the MMC was seeded with activated sludge from an industrial WWTP treating the same wastewater used as organic feedstock. This unit consisted of a 4-L poly-methyl-methacrylate column-type reactor (700 mm of height, 60 of inner diameter) equipped with an internal riser according to a sequencing-batch-airlift-reactor (SBAR) configuration [23]. This reactor operated aerobically under the typical feast/famine (F/f) regime. Continuous aeration (3.5 LPM, corresponding to 2.4 cm sec$^{-1}$ of flux velocity) was provided by an air stone diffuser placed at the bottom of the reactor and coaxial with the riser, connected to an air blower.

The raw citrus wastewater was stored in a refrigerated storage tank, and after pH neutralization by sodium hydroxide, it was fed to the fermenter. From this, wastewater was fed in upflow mode to the reactor by means of a peristaltic pump. The effluent was discharged at different heights to modify the volumetric exchange ratio (VER) according to operational needs. All electrical devices were connected to a programmable logic controller (PLC) that handled cycle operations. The AGS was operated in cycles of 12 h, consisting of 10 min of influent feeding, 690 min of aeration, 5 min of settling, 10 min of effluent discharge and lastly, 5 min of idle. The volume of wastewater fed and discharged in each cycle changed according to the operating period (see Section 2.3), thus resulting in different hydraulic retention times (HRT). An additional stream containing nitrogen and phosphorus (2 gCH$_4$N$_2$O L$^{-1}$ and 1 gK$_2$HPO$_4$ L$^{-1}$) was added at the beginning of the cycle to obtain a ratio between carbon (as total chemical oxygen demand—COD)/nitrogen/phosphorous equal to 100: 5: 1.

Another enrichment SBR (22 L) with flocculent sludge was operated in parallel with the AGS. The SBR operated under the same conditions as the AGS, apart from the settling time, which was set to 2 hours. For additional information about the SBR, the reader is referred to the literature [24].

The storage capacity of the enriched MMC was assessed by carrying out several accumulation assays. These were performed in a 1.5 L FBR seeded with the sludge of both the AGS and SBR, in which an automatic dissolved oxygen control system was provided for the calculation of the oxygen uptake rate (OUR) in order to monitor the accumulation capacity of the MMC.

Figure 1 shows the plant layout.

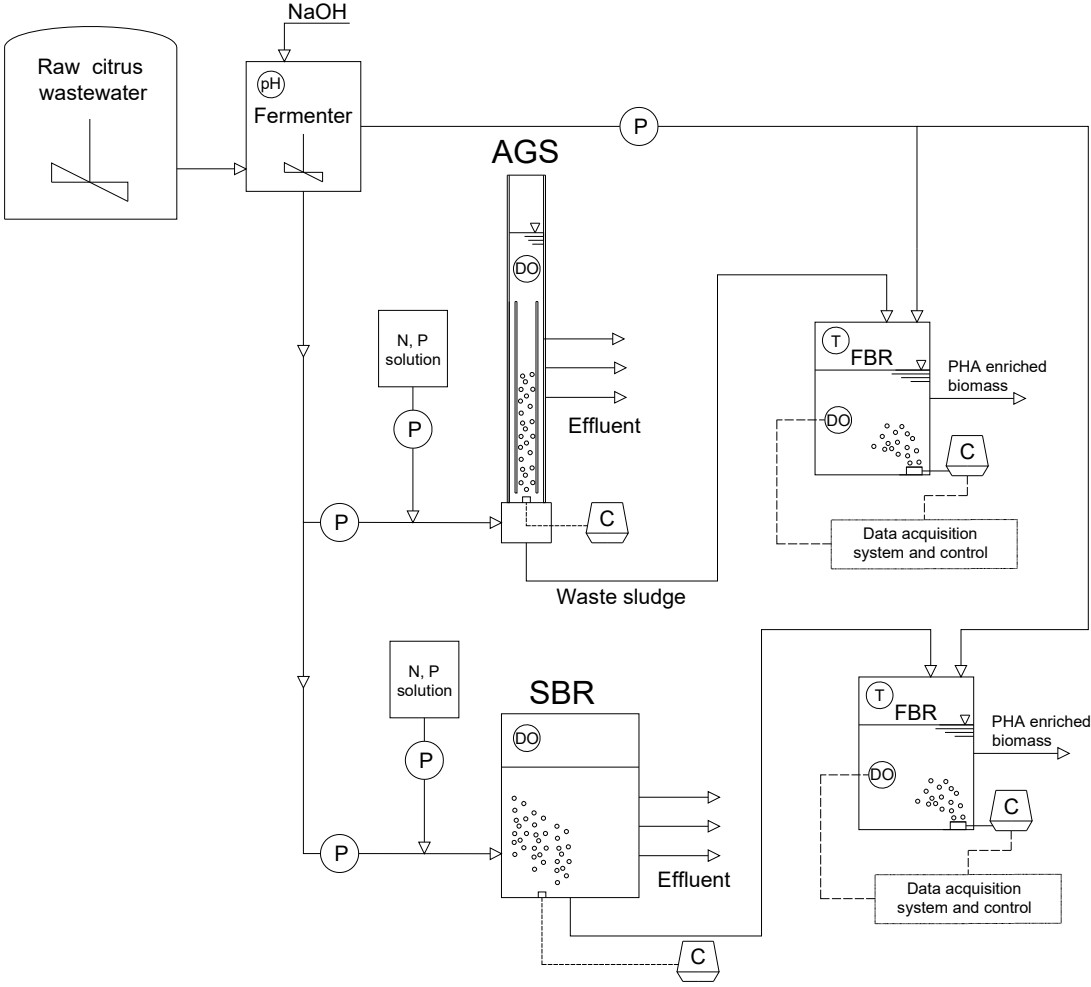

**Figure 1.** Pilot plant layout. Legend: C: air blower; P: pump; pH: pH controller; DO, dissolved oxygen sensor; T: temperature sensor.

### 2.3. Operational Strategy of the Enrichment Reactors

The enrichment AGS was operated for 204 days divided into three periods, namely Period 1 (P1), Period 2 (P2) and Period 3 (P3), having a duration of 126, 37 and 41 days, respectively. The experimental periods were characterized by three different OLRs, equal to 1 kgCOD m$^{-3}$d$^{-1}$ (P1), 2 kgCOD m$^{-3}$d$^{-1}$ (P2) and 3 kgCOD m$^{-3}$d$^{-1}$ (P3), obtained by modifying the VER of the enrichment reactor.

Period 1 was aimed at achieving complete granulation of the flocculent-activated sludge used as inoculum. Aerobic granules were obtained after 70 days. Hereafter a period equal to 3 times the sludge retention time (SRT), the OLR was increased, and Period 2 started. All the other periods lasted at least a period equal to 3 SRTs to obtain a steady state. The total suspended solids (TSS) concentration was maintained at approximately $5.0 \pm 0.4$ gTSS L$^{-1}$ by daily purging the excess sludge according to the biomass growth yield. Accordingly, the food-to-microorganisms ratio (F/M) resulted equal to 0.20 kgCOD kgTSS$^{-1}$d$^{-1}$ during Period 1, 0.41 kgCOD kgTSS$^{-1}$d$^{-1}$ during Period 2 and 0.64 kg COD kgTSS$^{-1}$d$^{-1}$ during Period 3, on average.

The operating conditions of the 2 enrichment reactors are reported in Table 2.

**Table 2.** Main operating parameters of the enrichment AGS and SBR during the experiment.

| | P1 | | P2 | | P3 | |
| | AGS | SBR | AGS | SBR | AGS | SBR |
|---|---|---|---|---|---|---|
| Duration [d] | 126 | 78 | 37 | 32 | 41 | 62 |
| Daily flow [L d$^{-1}$] | 0.90 | 5 | 1.80 | 10 | 2.72 | 15 |
| VER [-] | 0.11 | 0.11 | 0.22 | 0.23 | 0.33 | 0.34 |
| Biomass concentration [gTSS L$^{-1}$] | 4.89 ± 0.21 | 4.56 ± 0.12 | 5.11 ± 0.09 | 4.42 ± 0.09 | 5.06 ± 0.11 | 4.46 ± 0.15 |
| OLR [kgCOD m$^{-3}$d$^{-1}$] | 1.02 ± 0.06 | 1.07 ± 0.03 | 2.08 ± 0.04 | 2.04 ± 0.09 | 3.12 ± 0.08 | 3.05 ± 0.15 |
| F/M [kgCOD kgTSS$^{-1}$d$^{-1}$] | 0.20 ± 0.03 | 0.22 ± 0.08 | 0.41 ± 0.06 | 0.43 ± 0.04 | 0.62 ± 0.08 | 0.63 ± 0.05 |
| SRT [d] | 23 ± 1 | 22 ± 2 | 10 ± 3 | 7.0 ± 1.5 | 9 ± 1.6 | 5.1 ± 0.9 |

Legend: VER (volumetric exchange ratio); OLR (organic loading rate); F/M (food-to-microorganism ratio); SRT (sludge retention time).

At the end of each period, at least 2 accumulation assays were performed in order to evaluate the maximum EPS and PHA storage yields achievable by the enriched granular and flocculent sludges.

The accumulation reactor was operated aerobically by maintaining the dissolved oxygen (DO) concentration between 2–4 mg L$^{-1}$ and according to the aerobic dynamic feeding (ADF) regime [25]. This was obtained by adding the substrate (fermented citrus wastewater) in pulses of 50 mL. To maximize the biopolymers accumulation and prevent biomass growth, no nutrient source was fed. The maximum accumulation capacity was assumed when the DO remained close to the saturation value after adding a substrate pulse.

### 2.4. Analytical Methods

All the physical-chemical analyses for the assessment of total suspended solids (TSS), volatile suspended solids (VSS), Chemical Oxygen Demand (COD), total nitrogen (TN) and total phosphorus (TP) were carried out according to the standard methods [26].

COD measurements were performed on both the influent and effluent samples of the enrichment AGS and SBR reactors without any pretreatment in order to assess the systems' purification performances. The settling properties of the granular sludges in the enrichment reactors were assessed by calculating the SVI$_{30}$ and SVI$_5$. These were calculated by dividing the volume occupied by the sludge inside a 1 L graduated cylinder after 30 min, or 5 min in the case of SVI$_5$, of static settling by the concentration of TSS in the sample. A unit ratio between SVI$_5$ and SVI$_{30}$ was considered as an indicator to assume the achievement of complete granulation in the enrichment AGS. Moreover, the average particle size and the particle size distribution (PSD) of granular sludge were measured by an optical granulometer (QICPIC—Sympatec). The percentage of granules in the enrichment AGS was assumed to be equal to the percentage of particles with a size greater than 400 μm [27].

PHA and EPS were measured in the FBR at the beginning of the assay, and after that, the maximum accumulation capacity was achieved. At the same time, TSS, VSS and COD were measured in the same samples to assess the carbon mass balance. Specifically, PHAs were extracted from the sludge samples by applying the procedure suggested by Fiorese and co-authors [28] using 1–2 propylene carbonate as solvent. The purity of the extracted PHA was measured by a spectrophotometer using commercial standards as blanks (PHB-HV 88–12% and PHB by Sigma Aldrich). EPSs were extracted according to the heating method [29]. Then, the extracted EPSs were characterized by measuring the protein [30] and carbohydrate concentrations [31] using bovine serum albumin and glucose as standards, respectively.

DO, pH and temperature measurements in the enrichment and accumulation reactors were performed by online sensors (WTW). Table 3 summarizes the average values of DO, pH and temperature measured in the enrichment reactors during the experimental periods.

**Table 3.** Average DO, pH and temperature in the enrichment AGS and SBR reactors.

| | P1 | | P2 | | P3 | |
|---|---|---|---|---|---|---|
| | **AGS** | **SBR** | **AGS** | **SBR** | **AGS** | **SBR** |
| DO [mgO$_2$ L$^{-1}$] | 4.32 | 4.96 | 3.84 | 4.12 | 3.68 | 3.91 |
| pH [-] | 8.12 | 8.03 | 7.99 | 8.31 | 8.19 | 8.27 |
| T [°C] | 21.3 | 21.5 | 22.7 | 22.5 | 24.1 | 23.9 |

Data referred to average values during the entire period.

*2.5. Calculation*

The percentage of biopolymers produced (EPS or PHA) expressed in dry weight (%wt) was calculated by dividing the mass of PHA and EPS by the mass of VSS present in the medium (Equation (1) and (2)):

$$PHA(\%) = \frac{gPHA}{gVSS} \times 100 \tag{1}$$

$$EPS(\%) = \frac{gEPS}{gVSS} \times 100 \tag{2}$$

The mass balance for organic carbon (as total COD) in FBR was assessed according to Equation (3):

$$COD_{d(gCOD)} = PHA_{p(gCOD)} + EPS_{p(gCOD)} + X_{p(gCOD)} + COD_{r(gCOD)} \tag{3}$$

- COD$_d$: total mass (g) of COD dosed during the entire accumulation assay until the maximum accumulation capacity was obtained;
- PHA$_p$: mass of PHA produced (g), obtained as the difference between the final PHA mass at the end of the accumulation assay and that measured at the beginning. The following stoichiometric coefficients were assumed for referring PHA mass as COD (1.67 gCOD gPHB$^{-1}$ e 1.92 gCOD gPHV$^{-1}$);
- EPS$_p$: the mass of EPS (g) produced, which was obtained as the sum of the proteins (PN) and carbohydrates (PS) produced during the assay multiplied by the respective stoichiometric coefficients obtained experimentally (1.36 gCOD gPS$^{-1}$ e 1.40 gCOD gPN$^{-1}$);
- COD$_r$: the residual mass of COD obtained from the product of the reactor volume by the COD concentration measured in the supernatant at the end of the accumulation assay;
- X$_p$: the new biomass produced during the assay, measured as the volatile suspended solids produced during the accumulation assay as COD (1.42 gCOD gVSS$^{-1}$), minus the mass of biopolymers (EPS+PHA) as COD and the residual COD at the end of the assay.

**3. Results**

*3.1. Characteristics of Aerobic Granular Sludge*

The granulation process was monitored by means of the PSD and the ratio of SVI$_5$/SVI$_{30}$. Figure 2 depicts the average PSD obtained in each experimental period and that of the seed sludge (Figure 2a), as well as the trend of the SVI$_5$/SVI$_{30}$ and SVI$_5$ (Figure 2b).

The seed sludge was characterized by an average floc size (particle size corresponding to 50% of the cumulative distribution) close to 100 μm and an SVI$_5$/SVI$_{30}$ close to 3. According to the SVI$_5$/SVI$_{30}$ ratio, complete granulation was obtained after 80 days, although the percentage of sludge particles assimilated to granules (>400 μm) was about 90%. This suggested the coexistence of granules and residual flocculent sludge in the bulk of the enrichment AGS reactor. At steady state, the SVI$_5$ was close to 40 mL gTSS$^{-1}$, while the average particle size of granules was 1.8 mm.

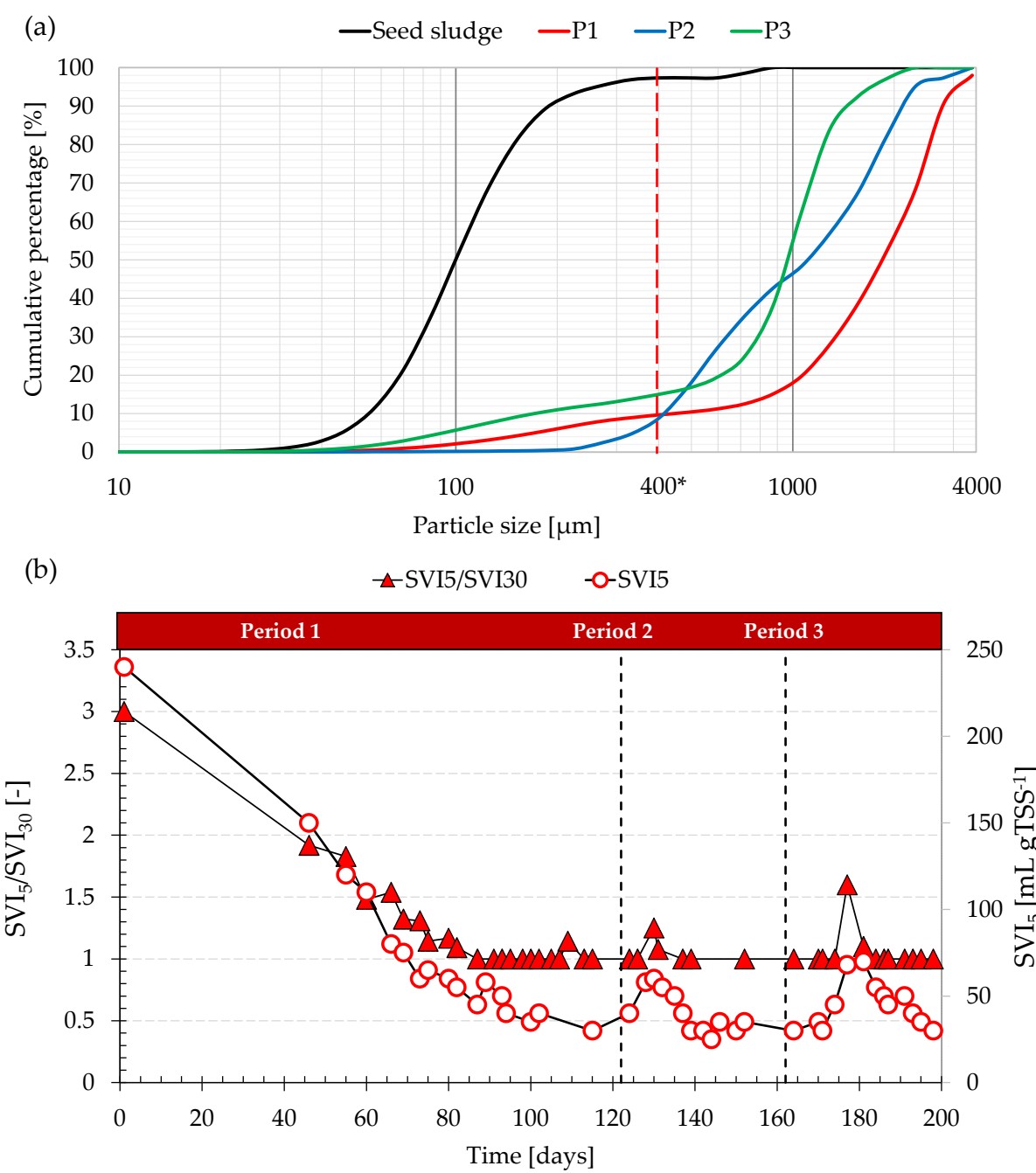

**Figure 2.** Particle size distribution of aerobic granular sludge during the experiment (**a**); trends of $SVI_5/SVI_{30}$ ratio and $SVI_5$ (**b**). 400 μm represents the cut-off value between flocculent and granular sludge assumed in this study.

In Period 2, the $SVI_5/SVI_{30}$ remained at approximately 1 during the entire period. Similarly, the $SVI_5$ was constantly close to 40 mL $gTSS^{-1}$. In contrast to the previous period, the average size of granules decreased to approximately 1.2 mm, although the percentage of granular sludge was similar (90%). A significative disappearance of flocculent sludge was noted, as suggested by the negligible percentage of particles with a size smaller than 300 μm.

In Period 3, the average size of granules still decreased, reaching a value of approximately 0.90 mm and suggesting the occurrence of degranulation. Contextually, the percentage of flocculent sludge slightly increased, thereby resulting in a small reduction of the percentage of granules to 85%. Nonetheless, apart from a short period around the

180th day, the $SVI_5/SVI_{30}$ was maintained at 1, and the $SVI_5$ was, on average, close to 50 mL gTSS$^{-1}$, indicating the excellent settling performances of AGS.

The achieved results were in line with previous literature in which increased instability and related deflocculation were observed when increasing the OLR [32]. Indeed, high OLRs resulted in fast granule formation, but the developed granules were unstable due to the large size resulting from excessive microbial growth. According to previous literature, the increase in substrate availability resulted in a prolonged feast phase and the following reduction of that of famine [32]. Consequently, bacteria with higher growth kinetic were dominant in the aerobic granules at the expense of slow-growing microorganisms. Moreover, since no anaerobic feeding was performed in the present study, it was assumed that the growth of fast-growing bacteria prevailed over slow-growing, thus resulting in an albeit modest granule instability when increasing the OLR [33].

### 3.2. COD Removal Performances in the Enrichment AGS and SBR Reactors

Figure 3 depicts the average COD removal efficiencies obtained in the enrichment reactors during the experiment.

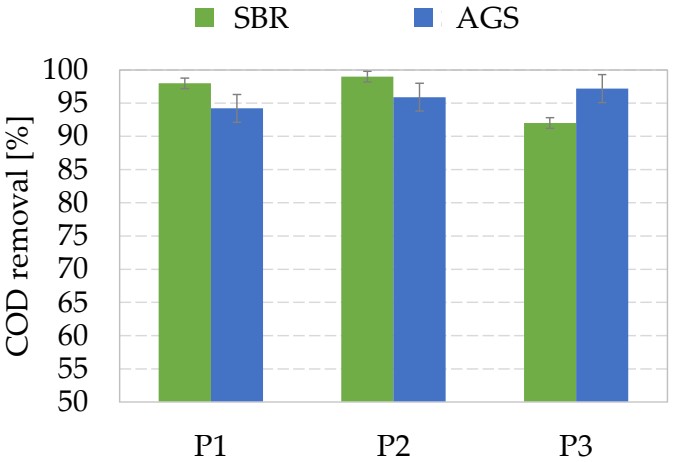

**Figure 3.** Average COD removal efficiencies in the enrichment ASG and SBR.

Both the enrichment AGS and SBR exhibited very high carbon removal, higher than 90% in general. In more detail, SBR showed a slightly decreasing trend with the OLR, although the effluent COD concentration resulted always lower than 400 mgO$_2$ L$^{-1}$, in compliance with EU regulations for industrial wastewater [34]. In contrast, AGS showed greater reliability in COD removal at increasing OLR. Indeed, even in Period 3, the average COD removal was higher than 95%, thereby resulting in effluent COD concentration lower than 200 mgO$_2$ L$^{-1}$.

In a previous study, it was pointed out that the simultaneous enrichment of MMC in biopolymers-storing populations via the feast/famine strategy and achievement of pollutant removal in compliance with regulations could be challenging in conventional activated sludge systems with flocculent biomass [24]. This was because, at high OLR, the occurrence of filamentous bulking decreased the sludge-settling properties, which worsened the effluent quality [35]. Therefore, this indicated that the enrichment and the wastewater treatment should be performed in different stages when using conventional systems, thereby involving an increase of the facilities required and plant footprint [36].

In the present study, it was noted that the MMC enrichment strategy did not cause a significant decrease in the AGS performances, thus suggesting that granular sludge systems could perform the enrichment and the wastewater treatment phases simultaneously. Moreover, the possibility to operate under higher OLR result in a further decrease of the treatment/enrichment reactor and of plant footprint as well.

### 3.3. Assessment of Biopolymers Accumulation Capacity

Accumulation assays were performed in the FBR once the enrichment AGS and SBR reached a steady state. More precisely, a ratio between the length of feast and famine phases lower than 0.20 was assumed as an indicator for the selection of microorganisms with storage capacity [37].

The operation of the enrichment AGS became stable after the 90th, 140th and 181st day in Period 1, Period 2 and Period 3, respectively. Thus, the accumulation assays were performed in two replicates on the 120th–121st (P1), 160th–161st (P2) and 123rd–124th days (P3).

Figure 4 shows the average maximum biopolymers content (as a sum of PHA and EPS) obtained from the batch assays in each period, in comparison with that obtained from the enriched SBR.

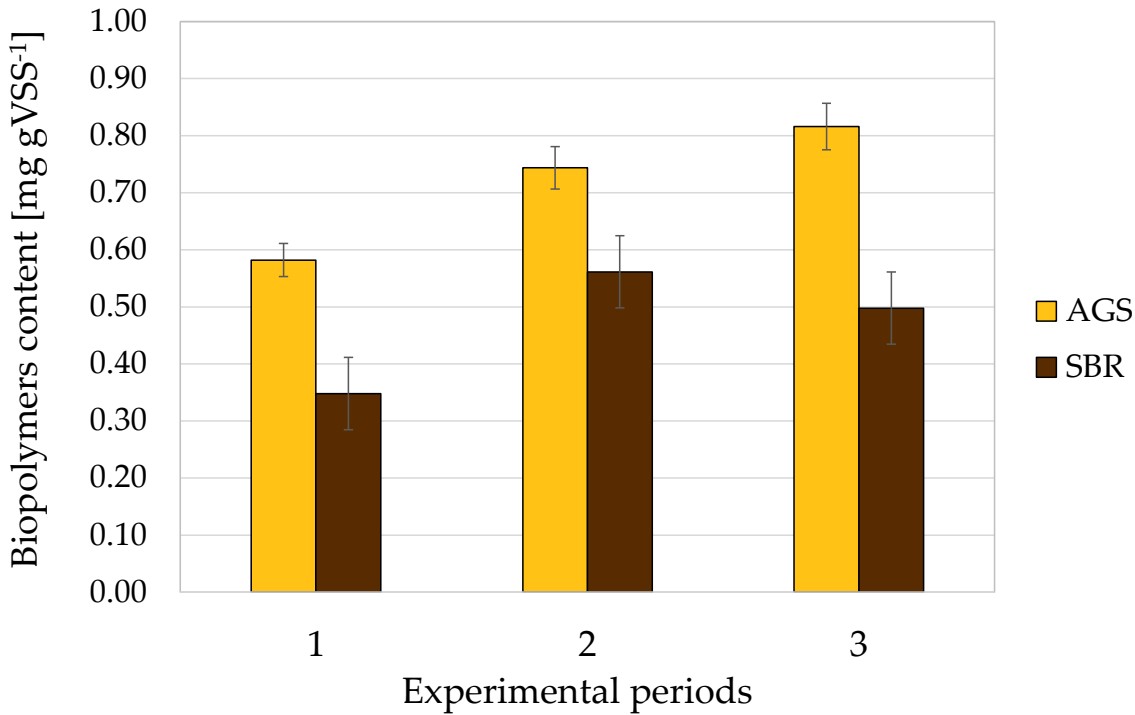

**Figure 4.** Average biopolymers content (as a sum of PHA and EPS) obtained at the end of the accumulation assays in the FBR from the AGS and SBR systems.

The biopolymer content referred to the unit of VSS which showed higher results in the AGS in all the periods. In Period 1, the maximum accumulation capacity in the AGS was close to 0.60 mg gVSS$^{-1}$, whereas, in the SBR, it was half (0.35 mg gVSS$^{-1}$). When increasing the OLR in the enrichment reactors, the MMC of AGS and SBR showed different behaviors. Indeed, while a positive linear relationship between the OLR in the enrichment reactor and the maximum biopolymer accumulation in the FBR was noted in the AGS, a decrease was observed in the SBR after a certain OLR. This result indicated that up to a certain OLR, the selection of biopolymers accumulating microorganisms in the enrichment reactor was less effective in the SBR, whereas this was maintained in the AGS, although it cannot be excluded the existence of such a threshold value also for the aerobic granular sludge systems.

In previous studies, it was already observed that too high substrate availability was not favorable to select microorganisms able to convert organic carbon into intracellular or extracellular storage polymers [38,39]. The feast/famine selection strategy is based on the capacity of the accumulating population to grow on the stored substrates produced during the feast phase when the external substrate is no longer available. This allows us to

wash out from the system those microorganisms not able to produce storage compounds, thereby resulting in the enrichment of the MMC with biopolymers storing populations [40]. Indeed, when organic substrate availability increases, the length of the feast phase is higher while reducing that of starvation. Consequently, microorganisms with internal storage capacity are no longer favored over non-storing populations during the famine phase. In AGS systems, because of the higher microbial density of the bio-aggregates with respect to the flocculent sludge in conventional SBR, substrate uptake during the feast phase is faster [41,42]. This, under the same operating conditions (e.g., cycle length, OLR, aeration rate, VER, etc.), could result in a longer famine phase, which guarantees a more effective selection of biopolymers-accumulating microorganisms. In the present study, it was noted that the length of the feast phase was shorter in the AGS than in the SBR in all the periods. Indeed, in Period 1, the feast phase lasted about 35 min at a steady state, whereas in the SBR, it was close to 45 min. Similarly, at a steady state in Period 2 and Period 3, the feast phase was shorter in the AGS, being equal to 45 min (P2) and 60 min (P3), respectively, whereas, in the SBR, it was 55 min (P2) and 75 min (P3) [24]. This result pointed out that the feast/famine selection strategy is more effective in AGS systems than conventional SBR with flocculent sludge.

Another possible reason to explain the higher biopolymer accumulation in the AGS could be related to the better settling properties of granular sludge. In other studies, it was reported that high OLR caused the worsening of the sludge settleability and wash-out of polymers-accumulating microorganisms occurred [43]. More precisely, the overgrowth of filamentous bacteria was noted in the SBR when the OLR increased, and this was associated with a slight loss in polymer accumulation capacity by the MMC. For further details on the SBR, the reader is referred to the literature [24]. In the present study, the improvement of sludge settling properties due to aerobic granulation enabled us to avoid significant losses in biopolymer accumulation capacity as observed in the enrichment reactor with flocculent sludge. Overall, AGS has proved to be more efficient than flocculent sludge for biopolymer recovery from wastewater treatment.

*3.4. Biopolymers Composition*

The results reported in the previous section showed the effectiveness of using AGS as an enrichment technology for biopolymer production. In Figure 5, the average compositions of the produced biopolymers in terms of EPS and PHA achieved in the FBR using the enriched AGS and SBR sludges are reported.

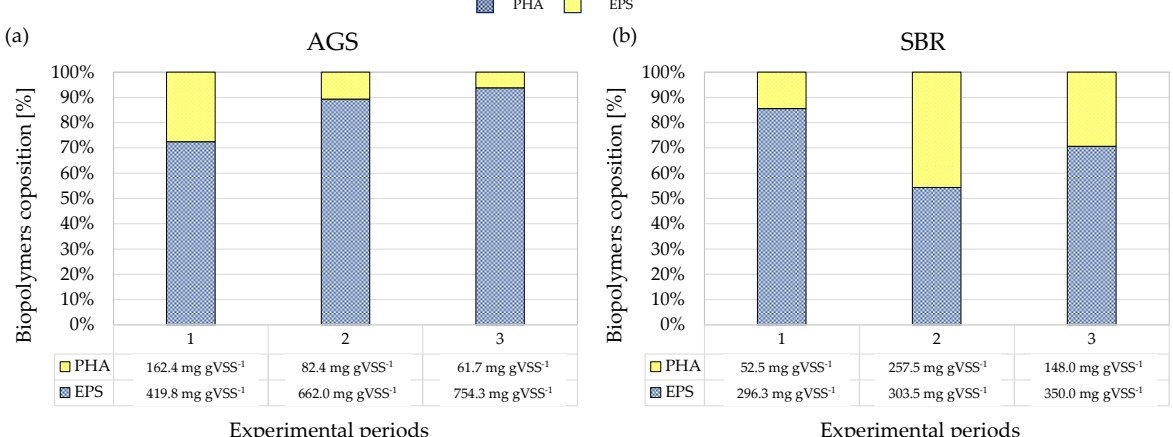

**Figure 5.** Percentages of PHA and EPS in the extracted biopolymers at the end of the accumulation assays performed in the FBR from the AGS (**a**) and SBR (**b**) systems.

In Period 1, the biopolymers extracted from the AGS were mainly constituted by EPSs, accounting for approximately 70% (419.8 mg gVSS$^{-1}$), whereas the PHA content was

almost 30% (162.4 mg gVSS$^{-1}$). A similar result was observed in the SBR. Indeed, EPSs were the main extracted biopolymer (83%, 296.3 mg gVSS$^{-1}$), and the PHA content was lower (17%, 52.5 mg gVSS$^{-1}$).

The abundance of EPSs in biopolymers from granular sludge significantly increased in Period 2 at the expense of PHA content. Indeed, the percentage of EPSs increased up to 90% (662 mg gVSS$^{-1}$), whereas that of PHA decreased to 10% (82.4 mg gVSS$^{-1}$) and by almost 50% with respect to the previous period (82.4 mg gVSS$^{-1}$ vs. 162.4 mg gVSS$^{-1}$). In contrast, in the SBR, the EPS and PHA contents were quite similar. Specifically, the EPS content was similar to that observed in the previous period (296 mg gVSS$^{-1}$ vs. 303 mg gVSS$^{-1}$), whereas the increase in the PHA content was statistically significant (52 mg gVSS$^{-1}$ vs. 257 mg gVSS$^{-1}$).

A similar tendency was noted in the AGS during Period 3. The percentage of EPS still increased up to 94% (754.3 mg gVSS$^{-1}$), whereas that of PHA reduced to 6%, although its reduction in terms of absolute value was lower than what was observed passing from Period 1 to Period 2 (82.4 mg gVSS$^{-1}$ vs. 61.7 mg gVSS$^{-1}$). In the SBR, a statistically significant increase in EPS content and a decrease in that of PHAs were observed.

Overall, the results obtained indicated that AGS enabled us to achieve a higher EPS yield than SBR independently of the OLR in the enrichment reactor. Indeed, the EPS content in the extracted biopolymers from granular sludge was twice that measured in the flocculent sludge. A previous study reported that the amount of EPS produced by AGS treating municipal wastewater was about 300 mg gVSS$^{-1}$ [44,45], although only a fraction of such polymers, namely structural EPSs (alginate-like exopolysaccharides), has a relevant market [46].

Referring to PHA, the results evidenced that conventional SBR enabled to obtain a higher PHA production than AGS. The average PHA content measured in the FBR of the SBR system was almost double that in the AGS, except for in Period 1. The results obtained in the SBR were in line with previous studies carried out with industrial agro-based wastewaters [47], whereas that of AGS pointed out a considerable gap with respect to the average values reported in the literature. Consequently, the enrichment stage of the process in the AGS system needed to be optimized to improve culture selection and maximize PHA content.

In this respect, it was observed that when increasing the OLR in the enrichment reactor, the capacity of the MMC to produce PHA noticeably decreased in favor of EPS. In contrast, operating under low OLR (<1 kgCOD m$^{-3}$d$^{-1}$) resulted in a better PHA yield than SBR, although the achieved value (162.4 mg gVSS$^{-1}$) was lower than the reference ones reported in the literature [48]. Concerning the AGS technology, there are still few published studies on the impact of operating conditions on PHA synthesis. Generally, high OLR (>3 kgCOD m$^{-3}$d$^{-1}$) is reported to be beneficial to maximize PHA production in AGS systems, whereas low OLR (<2 kgCOD m$^{-3}$d$^{-1}$) is more indicated for conventional activated sludge [49]. Indeed, the effect of OLR on activated sludge was confirmed in this study, whereas opposite results were obtained referring to the AGS. A possible reason could be associated with substrate diffusion resistance observed in AGS, as the protection created by the AGS structure limited carbon concentration within these clusters [50]. Thus, a possible substrate limitation could be the reason for the lower PHA productivity observed in the AGS.

### 3.5. Analysis of Carbon Utilization in the FBRs

To examine how the organic matter was used by the enriched MMC of the AGS and SBR in the FBRs, mass balances were assessed to evaluate the conversion of the COD into intracellular biopolymers (PHA), extracellular biopolymers (EPS) and new biomass. All these elements were expressed in terms of COD using the respective conversion coefficients reported in Section 2.5. The achieved results are shown in Figure 6.

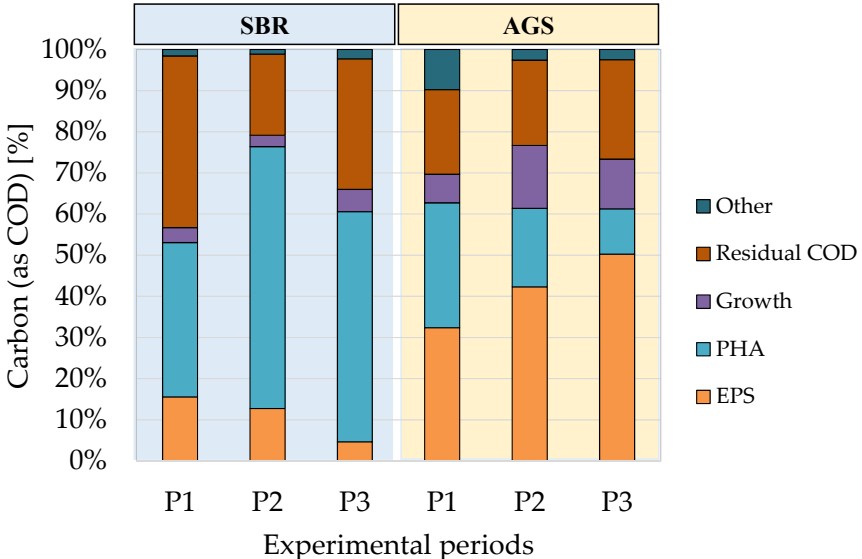

**Figure 6.** COD mass balance results over the three periods.

The enriched MMC of the SBR and AGS showed different behavior in terms of carbon channeling into PHA and EPS. The organic carbon available for the MMC in the FBRs was mainly driven toward PHA in that of SBR and EPS in the AGS. More precisely, a maximum PHA storage yield close to 70% (gPHA$_{(as COD)}$/gCOD) was obtained in the SBR when the enrichment reactor was operated with an OLR of 2 kgCOD m$^{-3}$d$^{-1}$, whereas the maximum value obtained in the AGS was slightly lower than 30% in Period 1. In contrast, more than 50% (gEPS$_{(as COD)}$/gCOD) of the COD was used to synthesize EPS in the AGS in Period 3 (maximum value), whereas only 15% was the maximum EPS yield obtained in the SBR (Period 1).

A clear relationship between the fraction of COD channeled toward EPSs and the OLR in the enrichment reactor was observed. When the OLR increased, the mass of COD used to synthesize EPS increased in the AGS (30–50%), whereas it decreased in the SBR (15%–5%). An opposite trend referring to PHA was noted in the AGS, where COD driven toward PHA synthesis decreased from 30% to 11% as the OLR increased.

Referring to the other elements of the mass balance, any differences statistically significant were found. Although a slightly higher yield referred to new biomass synthesis was found in the AGS, the ANOVA test did not highlight any statistically significant difference among the results ($p$-value > 0.05). Similarly, the residual COD was slightly lower in the FBR of the AGS system, although even in this case, no statistical significance was observed.

The above results clearly indicated that MMC enrichment in the SBR was more favorable to the selection of the PHA-storing population, and the OLR constitutes a key-control parameter to maximize the process yield. On the other hand, the enrichment of MMC using AGS by applying similar operating strategies to conventional SBR led to the selective enrichment of EPS-storing microorganisms. This favored metabolic pathways for storing organic carbon extracellularly rather than intracellularly. The different behavior was more evident when operating at a higher OLR.

A possible reason to explain why carbon was preferentially channeled toward EPS in AGS could be a side effect of the higher hydraulic selection pressure and shear forces applied in the AGS system to promote aerobic granulation. A short settling time and high aeration rate corresponding to high shear stress are well recognized as key factors for the transition from flocculent-activated sludge to aerobic granules [51]. Several studies in the past decades have demonstrated that an increase in both the hydraulic selection pressure and shear forces determined a noticeable excess of EPS secretion by bacteria [52]. Specifically, it was demonstrated that such metabolic stressors activated quorum-sensing signaling among microorganisms and a positive correlation between the signal activity

and EPS production was found [53]. Therefore, when AGS operated in the presence of metabolic stressors, it was likely that carbon was channeling toward EPS formation as the key element to enable microbial aggregation to withstand external disturbances.

In conclusion, the results obtained in this study suggested that the production of EPS from AGS could be considered the most effective process for sludge valorization through the biopolymers recovery pathway. Contrarily, PHA production is currently more convenient in conventional SBR. It could also be stressed that PHA production was so far optimized for activated sludge systems rather than granular sludge. Consequently, to maintain the same effectiveness, the operation of AGS would be adequately modified. Therefore, the effect of other operational parameters that could drive microbial metabolism toward the preferential production of EPS of PHA without affecting the granulation process should be investigated in future studies.

In particular, the effect of microbial community composition (the prevalence of glycogen accumulating or phosphate accumulating organisms), or the role of the feeding strategy (full aerobic, upflow-anaerobic, etc.), as well as the shear forces and hydraulic selection pressure should be elucidated.

*3.6. Potential Social and Economic Impact of Biopolymer Recovery from Citrus Wastewater*

Food waste industries produce many wastes during primary production and processing. The citrus industry also plays an important role in the agro-industrial sector, especially in Mediterranean countries such as Italy and Spain. The citrus industry produces large amounts of CPWW with high polluting potential, which management determines severe environmental and economic constraints for the citrus processing industry. Sustainable residues management, which entails technological innovation based on environmental and economic issues, sustainability, and economic resilience, has the potential to significantly increase the competitiveness of the citrus business. In this framework, this study demonstrated that a sustainable process could be implemented to cope with environmental regulations for effluent quality and for the purpose of applying circularity concepts to waste streams. Therefore, agroindustries may be able to use circularity as a strategy to achieve zero waste production, good water/energy/residue cycling, reduced environmental impact, generation of clean energy, and increased bioeconomy.

## 4. Conclusions

This study evaluated the feasibility of using AGS as an enrichment reactor for biopolymers production by MMC. The obtained results indicated that AGS enabled us to obtain a higher accumulation capacity with respect to conventional activated sludge in flocculent form. The maximum biopolymers accumulation capacity was close to 0.60 mgPHA-EPS gVSS$^{-1}$ in the AGS operating with OLR of 3 kgCODm$^{-3}$d$^{-1}$, whereas, in the SBR, it was about half (0.35 mgPHA-EPS gVSS$^{-1}$). Biopolymers extracted from the AGS were mainly constituted by EPSs (>70%), whose percentage increased up to 95% with the maximum OLR applied in the enrichment reactor. Thus, organic carbon was mainly channeled toward metabolic pathways for extracellular storing, resulting in a significant prevalence of EPS in the extracted biopolymers. In contrast, SBR enabled us to obtain a higher PHA production (50% of the biopolymers). The presence of metabolic stressors (e.g., hydraulic selection pressure, shear forces) as key factors for promoting aerobic granulation was supposed as the main force for organic carbon channeling toward EPSs rather than PHAs. In contrast, flocculent sludge was more favorable to the selection of the PHA-storing population, and the OLR applied in the enrichment reactor was found to be a key operating factor in driving the process toward PHA recovery. Based on the achieved results, it was concluded that AGS could be considered a suitable enrichment technology for EPS recovery from excess valorization. Future studies are necessary to optimize PHA productivity by AGS, focusing on the role of the main operating parameter affecting aerobic granulation.

**Author Contributions:** Conceptualization, S.F.C. and F.T.; methodology, S.F.C. and F.T.; software, S.F.C. and F.T.; validation, S.F.C., M.T. and G.V.; formal analysis, S.F.C. and F.T.; investigation, S.F.C.; resources, M.T., G.V.; data curation, S.F.C. and F.T.; writing—original draft preparation, S.F.C. and F.T.; writing—review and editing, M.T. and G.V.; visualization, M.T. and G.V.; supervision, M.T. and G.V.; project administration, M.T. and G.V.; funding acquisition, M.T and G.V. All authors have read and agreed to the published version of the manuscript.

**Funding:** This research was funded by the Ministry of Education, University and Research (MUR, Italy)—Project PON Ricerca e Innovazione 2014–2020—FSE REACT-EU—Azione IV (D.M. 1061/2021) and Azione VI (D.M. 1062/2021).

**Data Availability Statement:** Data will be available on request to the corresponding author.

**Acknowledgments:** Authors thank the "Agrumaria Corleone S.p.A." (Palermo) for the precious technical support. Furthermore, the authors warmly thank Alessia Sola for her valuable contribution during pilot plant operations.

**Conflicts of Interest:** The authors declare no conflict of interest.

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
