# Peer review of "Biopolymer Recovery from Aerobic Granular Sludge and Conventional Flocculent Sludge in Treating Industrial Wastewater: Preliminary Analysis of Different Carbon Routes for Organic Carbon Utilization"

_water, doi:10.3390/w15010047_

Round 1
Reviewer 1 Report
The article submitted for review presents the study evaluated the feasibility to use AGS as enrichment reactor for biopolymers production by MMC. The obtained results indicated that AGS enabled to obtain higher accumulation capacity respect to conventional activated sludge in flocculent form. Based on the achieved results, it was concluded that AGS could be considered as a suitable enrichment technology for EPS recovery from excess valorization. Future studies are necessary to optimize PHA productivity by AGS focusing on the role of the main operating parameter affecting aerobic granulation. I encourage the authors to continue this research. I was pleased to read the description of the conducted research.
In my opinion, a proper review of the literature has been made, which is an excellent introduction to the subject. The whole experiment, in my opinion, was well planned and described. However, the article needs a few corrections:
1. In chapter 2.1 there is table 2. in the article there is no table 1. The text mentions table 1. In chapter 2.3 another table 2. This needs to be corrected.
2. Please also make table descriptions in accordance with the guidelines for authors.
3. In my opinion, it would be worth introducing microbiological analysis into the characteristics of sewage. Please consider it.
This is very interesting research that should be continued. Thank you for considering my opinion.
Author Response
Reviewer#1
The article submitted for review presents the study evaluated the feasibility to use AGS as enrichment reactor for biopolymers production by MMC. The obtained results indicated that AGS enabled to obtain higher accumulation capacity respect to conventional activated sludge in flocculent form. Based on the achieved results, it was concluded that AGS could be considered as a suitable enrichment technology for EPS recovery from excess valorization. Future studies are necessary to optimize PHA productivity by AGS focusing on the role of the main operating parameter affecting aerobic granulation. I encourage the authors to continue this research. I was pleased to read the description of the conducted research.
In my opinion, a proper review of the literature has been made, which is an excellent introduction to the subject. The whole experiment, in my opinion, was well planned and described. However, the article needs a few corrections:
Authors response: Authors thank the Reviewer#1 for the interesting suggestions useful for improving the quality of the paper. The Authors accepted all the suggestions and modified the manuscript accordingly to the Reviewer’s comments.
Q1: 1. In chapter 2.1 there is table 2. in the article there is no table 1. The text mentions table 1. In chapter 2.3 another table 2. This needs to be corrected.
Authors response: The Authors thank the Reviewer#1 for the suggestions. We apologize for the mistake. Tables were numbered correctly in the revised version of the manuscript.
Q2: 2. Please also make table descriptions in accordance with the guidelines for authors.
Authors response: The Tables description was modified according to the Reviewer’s suggestion.
Q3: 3. In my opinion, it would be worth introducing microbiological analysis into the characteristics of sewage. Please consider it.
Authors response: The Authors thanks the Reviewer for the precious suggestion. We understand the importance of microbiological analysis in order to assess the effect of the MMC composition on the biopolymer accumulation yield and to tackle the competition between EPS and PHA-storing population. This aspect will be deepened in future studies.
This is very interesting research that should be continued. Thank you for considering my opinion.
Reviewer 2 Report
1. English needs MAJOR REVISION. It is highly recommended to submit this manuscript to a professional proofreader to improve the incorrect sentence structures and words that can lead to misleading information.
2. Revise the abstract and reduce general content.
3. Include a statement of novelty in the introduction.
4. Break down the paragraphs.
5. Maintain consistency in reporting the units.
6. Compare the findings in this study with the reported literature.
7. Please discuss the industrial and societal relevance of the study.
8. Conclusions should be more specific with data or new inferences generated in this research.
9. Add more relevant references to the paper. Chemosphere 280 (2021) 130595, Desalin Water Treat 68 (2017) 245-266, Journal of Industrial and Engineering Chemistry 60 (2018) 307-320, IET nanobiotechnology 10 (4) (2016) 244-253, IET Nanobiotechnology 11 (6) (2017) 746-753
Author Response
Reviewer#2
Authors’ response: Authors thank the Reviewer#2 for the comments. The Authors accepted the Reviewer’s suggestions and modified the manuscript accordingly.
Q1: 1. English needs MAJOR REVISION. It is highly recommended to submit this manuscript to a professional proofreader to improve the incorrect sentence structures and words that can lead to misleading information.
Authors’ response: The manuscript was carefully revised in order to improve the english quality.
Q2: 2. Revise the abstract and reduce general content.
Authors’ response: The abstract was revised and reduced according to the Reviewer’s suggestion.
Q3: 3. Include a statement of novelty in the introduction.
Authors’ response: The novelty of the study was better elucidated in the introduction.
Q4: Break down the paragraphs.
Authors’ response: Edited ad suggested.
Q5: 5. Maintain consistency in reporting the units.
Authors’ response: Edited ad suggested.
Q6: 6. Compare the findings in this study with the reported literature.
Authors’ response: A comprehensive comparison between the main findings of this study and the available literature was carried out. Nevertheless, the Authors point out that given the specificity of the study, a direct comparison with other studies was difficult to made and we referred to studies dealing with similar wastewater of this study.
Q7: Please discuss the industrial and societal relevance of the study.
Authors’ response: The industrial and social relevance of the study was briefly discussed in a new paragraph in the revised version of the manuscript.
Q8: 8. Conclusions should be more specific with data or new inferences generated in this research.
Authors’ response: More specific data were added in the conclusion section according to the Reviewer’s suggestion.
Q9: 9. Add more relevant references to the paper. Chemosphere 280 (2021) 130595, Desalin Water Treat 68 (2017) 245-266, Journal of Industrial and Engineering Chemistry 60 (2018) 307-320, IET nanobiotechnology 10 (4) (2016) 244-253, IET Nanobiotechnology 11 (6) (2017) 746-753
Authors’ response: The Authors thanks the Reviewer#2 for the suggestion. We added some of the suggested paper by selecting the more appropriate to the research topic.
Reviewer 3 Report
GENERAL REMARKS
The paper focuses on the recovery of high-value polymers from industrial sewage sludge in a circular economy context. The authors compare two different processes (conventional SBR and aerobic granular sludge) by means a laboratory scale experimentation.
They calculate the amount of PAH and EPS produced under different operation conditions.
The research design is rigorous and the results are clearly and logically explained.
The findings can provide valuable (and novel) information to the scientific community about the recovery feasibility of materials from the sewage of citrus production (of outstanding importance in the Mediterranean, also with a view to the European policy on Circular Economy). The quantification of PAH and EPS produced will allow the comparison with other processes.
I would encourage the authors to extend the experimentation to a greater scale and to investigate the microbial populations involved in biopolymers accumulation.
DETAILED REMARKS
Title: since the wastewater treatment plants are two (namely, a conventional SBR and an aerobic granular sludge, the title should be modified to better describe the case studies.
Lines 17-20 the measurement units of the polymers should be specified.
Line 21: replace which with whose.
Line 80: sp. must not be written in italic
Line 106: replace “the machineries” with “machinery”
Line 125: specify the reagent used for pH neutralization
Table 2: I suggest reporting the duration of each period (although written in the text) in brackets after each abbreviation (P1, P2, P3)
Line 172: please, add pH, DO and temperature values (possibly adding Figures/Tables in the supplementary material section)
Lines 253-254: rephrase the second part of this sentence; “was…90%”.
Lines 273-275: this interpretation makes sense, and it would be useful to add some hypothesis on the factors that might have favoured the growth of fast-growing bacteria during this period.
Lines 292-294: I would use the past tense “decreased”, referring to the specific study reported, because filamentous bulking occurrence is not always linked to OLR. Furthermore, in the presence of excess EPS, zoogleal bulking might occur preferably than filamentous one.
Author Response
Reviewer#3
GENERAL REMARKS
The paper focuses on the recovery of high-value polymers from industrial sewage sludge in a circular economy context. The authors compare two different processes (conventional SBR and aerobic granular sludge) by means a laboratory scale experimentation.
They calculate the amount of PAH and EPS produced under different operation conditions.
The research design is rigorous and the results are clearly and logically explained.
The findings can provide valuable (and novel) information to the scientific community about the recovery feasibility of materials from the sewage of citrus production (of outstanding importance in the Mediterranean, also with a view to the European policy on Circular Economy). The quantification of PAH and EPS produced will allow the comparison with other processes.
I would encourage the authors to extend the experimentation to a greater scale and to investigate the microbial populations involved in biopolymers accumulation.
Authors’ response: Authors thank the Reviewer#3 for the interesting comments suggestion for improving the paper. The Authors accepted all the suggestions and modified the manuscript accordingly. Moreover, the Authors thanks the Reviewer#3 for suggesting to extend the experimentation and certainly the Authors will consider this in future researches.
DETAILED REMARKS
Q1: Title: since the wastewater treatment plants are two (namely, a conventional SBR and an aerobic granular sludge, the title should be modified to better describe the case studies.
Authors’ response: The title was changed to “Biopolymers recovery from aerobic granular sludge and conventional flocculent sludge treating industrial wastewater: preliminary analysis of different carbon routes for organic carbon utilization” according to the Reviewer’s suggestion
Q2: Lines 17-20 the measurement units of the polymers should be specified.
Authors’ response: The measurement unit was specified as suggested.
Q3: Line 21: replace which with whose.
Authors’ response: Edited as suggested
Q4: Line 80: sp. must not be written in italic
Authors’ response: Edited as suggested
Q5: Line 106: replace “the machineries” with “machinery”
Authors’ response: Edited as suggested
Q6: Line 125: specify the reagent used for pH neutralization
Authors’ response: Sodium hydroxide was added for pH neutralization. The manuscript was edited accordingly.
Q7: Table 2: I suggest reporting the duration of each period (although written in the text) in brackets after each abbreviation (P1, P2, P3)
Authors’ response: The duration of each period was reported as suggested
Q8: Line 172: please, add pH, DO and temperature values (possibly adding Figures/Tables in the supplementary material section)
Authors’ response: A new table (Tab.3) reporting the average data of pH, DO and temperature was added in the revised manuscript.
Q9: Lines 253-254: rephrase the second part of this sentence; “was…90%”.
Authors’ response: The sentence was rephrased.
Q10: Lines 273-275: this interpretation makes sense, and it would be useful to add some hypothesis on the factors that might have favoured the growth of fast-growing bacteria during this period.
Authors’ response: A possible hypothesis supported by previous literature was added in the revised manuscript.
Q11: Lines 292-294: I would use the past tense “decreased”, referring to the specific study reported, because filamentous bulking occurrence is not always linked to OLR. Furthermore, in the presence of excess EPS, zoogleal bulking might occur preferably than filamentous one.
Authors’ response: The sentence was rephrased according to the Reviewer’s suggestion.